# ECM Substrates Impact RNAi Localization at Adherens Junctions of Colon Epithelial Cells

**DOI:** 10.3390/cells11233740

**Published:** 2022-11-23

**Authors:** Amanda C. Daulagala, Antonis Kourtidis

**Affiliations:** Department of Regenerative Medicine and Cell Biology, Medical University of South Carolina, Charleston, SC 29425, USA

**Keywords:** PLEKHA7, AGO2, DROSHA, E-cadherin, laminin, fibronectin, collagen, colon

## Abstract

The extracellular matrix (ECM) plays crucial roles in tissue homeostasis. Abnormalities in ECM composition are associated with pathological conditions, such as fibrosis and cancer. These ECM alterations are sensed by the epithelium and can influence its behavior through crosstalk with other mechanosensitive complexes, including the adherens junctions (AJs). We have previously shown that the AJs, through their component PLEKHA7, recruit the RNAi machinery to regulate miRNA levels and function. We have particularly shown that the junctional localization of RNAi components is critical for their function. Here, we investigated whether different ECM substrates can influence the junctional localization of RNAi complexes. To do this, we plated colon epithelial Caco2 cells on four key ECM substrates found in the colon under normal or pathogenic conditions, namely laminin, fibronectin, collagen I, and collagen IV, and we examined the subcellular distribution of PLEKHA7, and of the key RNAi components AGO2 and DROSHA. Fibronectin and collagen I negatively impacted the junctional localization of PLEKHA7, AGO2, and DROSHA when compared to laminin. Furthermore, fibronectin, collagen I, and collagen IV disrupted interactions of AGO2 and DROSHA with their essential partners GW182 and DGCR8, respectively, both at AJs and throughout the cell. Combinations of all substrates with fibronectin also negatively impacted junctional localization of PLEKHA7 and AGO2. Additionally, collagen I triggered accumulation of DROSHA at tri-cellular junctions, while both collagen I and collagen IV resulted in DROSHA accumulation at basal areas of cell–cell contact. Altogether, fibronectin and collagens I and IV, which are elevated in the stroma of fibrotic and cancerous tissues, altered localization patterns and disrupted complex formation of PLEKHA7 and RNAi components. Combined with our prior studies showing that apical junctional localization of the PLEKHA7-RNAi complex is critical for regulating tumor-suppressing miRNAs, this work points to a yet unstudied mechanism that could contribute to epithelial cell transformation.

## 1. Introduction

The extracellular matrix (ECM) is a dynamic meshwork that constantly re-adjusts to accommodate homeostasis of a healthy extracellular environment [1,2,3,4]. Generally, the ECM is composed of two compartments, namely the basement membrane and the interstitial matrix. The basement membrane is a thin layer in direct contact with the epithelium. It is mainly composed of collagen IV and laminin, which maintain the stability and structural integrity of the basement membrane. The interstitial matrix is a thicker layer located below the basement membrane, composed mainly of fibrillar collagens, fibronectin, and elastin, among other proteins [1,2,3,4]. Of the fibrillar collagens, collagen I is the most abundant in humans. It forms large fibers that fill a considerable space in the interstitial matrix, thus providing mechanical strength to the tissue [5,6,7]. Fibronectin is another widely studied component of the interstitial matrix that intermingles with other matrix components, including collagen I, to create an intricate network [8,9,10].

A tissue disruption, such as a wound, can induce new ECM synthesis and, eventually, remodeling. However, continuous insults, such as chronic inflammation, can promote constant remodeling and excessive accumulation of ECM, leading to a condition known as fibrosis, which accounts for most deaths in the modern world [11]. Prolonged fibrosis can lead to liver and kidney failures [12,13,14,15,16,17], scar tissue in the heart [18,19,20], or cancer [21,22,23]. Under pathological conditions, the ratio of ECM proteins is altered, primarily due to increased levels of collagen I, collagen IV, and fibronectin. In fact, elevated collagen and fibronectin are considered markers of tumor progression [24,25,26,27]. Clinical treatments have been developed to inhibit the effects of fibronectin in cancer, whereas current studies also target collagen IV as a potential therapeutic [28,29,30,31]. Overall, a critical role of the ECM in tumor progression has been broadly considered [25,32,33,34,35]. 

The ECM connects to cells via focal adhesions, which in turn crosstalk with other cellular mechanosensitive structures, such as the adherens junctions (AJs) [36,37,38]. As such, changes in the ECM can be sensed by AJs and vice versa. The AJs is a cell–cell adhesion complex, composed of cadherin and catenin family proteins, critical for maintaining epithelial integrity that has been involved in tumorigenesis and cancer progression [39,40,41,42,43,44,45,46,47]. Indeed, disruption of AJ integrity is a hallmark of epithelial cancers [48,49,50]. ECM alterations frequently observed in tumors [51,52,53] can negatively impact AJs [54,55,56,57,58,59,60]; however, whether ECM promotes pro-tumorigenic behavior through influencing certain changes in AJs, is still a matter of investigation.

Our previous studies have shown that PLEKHA7 (Pleckstrin homology domain-containing family A member 7), a protein associated with the E-cadherin—p120 catenin—based junctions, recruits the core components of the RNA interference (RNAi) machinery that regulate miRNA biogenesis and function, such as the microprocessor components DROSHA (Drosha Ribonuclease III) and DGCR8 (DiGeorge Syndrome Critical Region Gene 8), and the RNA-induced silencing complex (RISC) components Argonaute 2 (AGO2) and GW182 (Glycine-Tryptophan Protein Of 182 KDa), to mature apical AJs. This way, PLEKHA7 regulates the levels and activity of a set of miRNAs [61,62,63,64]. We have shown that PLEKHA7 and RNAi components are localized at apical AJs in normal colon tissues but are absent from areas of cell–cell contact in tumors [65]. We have also shown that loss of junctional localization of PLEKHA7 and RNAi components leads to decreased processing and activity of miRNAs, to oncogene upregulation, and to pro-tumorigenic cell behavior [62,63,64,65]. 

The colonic epithelium undergoes continuous remodeling and is also constantly exposed to prolonged insults in diseases, such as during inflammatory bowel disease, exposing it to fibrotic conditions [66]. In fact, the risk of colorectal cancer, which is the third most diagnosed and second deadliest form of the disease, increases by 50% under fibrotic conditions [67,68,69,70]. Indeed, colon epithelial cells are in an extensive crosstalk with the stromal ECM. For example, brush border enzyme activity levels, cell spreading rates, migration rates, and proliferation rates of colon epithelial Caco2 cells change when plated on different ECM substrates, such as laminin, fibronectin, collagen I, and collagen IV [71]. Importantly, during tumorigenesis, the composition of the colonic mucosa changes considerably and exhibits increased levels of fibronectin and collagen I, compared to normal conditions [24,28], whereas increased collagen IV in metastatic tumors is associated with poor survival [72,73,74,75]. Although it is well-documented that this switch in the ECM composition and the overall occurrence of fibrotic tissue correlates with colorectal cancer progression, the mechanistic underpinnings of this relationship are still poorly understood. Since the AJs respond to cues from the ECM, but also recruit and regulate the RNAi machinery to suppress pro-tumorigenic signals, we inquired in this work whether different ECM substrates could affect junctional localization of PLEKHA7 and of RNAi components, which we have shown is critical for AJ’s oncogene-suppressing function [62,63,65]. To investigate this, we cultured well-differentiated Caco2 colon epithelial cells on laminin, fibronectin, collagen I, and collagen IV, and examined changes in the junctional localization of PLEKHA7, DROSHA, and AGO2. Our results show that different ECM components can indeed impact localization and complex formation of RNAi at AJs, providing insights into a new mechanism that connects changes in ECM with intracellular RNAi regulation through the AJs, with potential implications in tumorigenesis.

## 2. Materials and Methods

### 2.1. Coverslip Coating

Calculations for working solutions were made according to the table below (Table 1), for coverslips 484 mm^2^ in area (22 mm × 22 mm in length/width—Denville Scientific Inc., Holliston, MA, USA, cat #M1100-01-ULTRA COVER glass, 0.14 mm thick), and for 250 µL total solution, per coverslip. Collagen I solution was purchased in working dilution and 1 mL of solution was added to each coverslip.

Coverslips were coated with fibronectin and laminin and kept in the cell culture hood for 45 min after application of diluted solutions. For fibronectin, excess fluid was removed, and the coated surface was rinsed 3 times with culture medium before plating the cells. For laminin, excess fluid was removed, and the coated surface was left to air-dry for 10 min before plating cells. Collagen I—coated coverslips were kept in the tissue culture incubator at 37 °C for 2 h after applying the diluted solution. Then, coated surfaces were rinsed twice with PBS before plating cells. Collagen IV—coated coverslips were left in the laminar flow hood for 2 h after applying the diluted solution. Then, coated surfaces were rinsed twice with PBS before plating cells. The concentration of each ECM component in the commercially available array (Advanced BioMatrix, Carlsbad, CA, USA; ECM Select^®^ Array Kit Ultra-36, cat #5170) was 250 µg/mL when individually, 125 µg/mL each when in combination with another component, 83.3 µg/mL each when in combination with two other components, and 62.5 µg/mL each when in combination with three other components, according to the vendor’s protocol [76].

### 2.2. Cell Culture

Caco2 cells (ATCC, Manassas, VA, USA, cat #HTB-37) were grown in MEM cell culture medium (Corning Inc.—Mediatech Inc., Manassas, VA, USA, cat #MT10010CV), supplemented with 10% FBS (Life Technologies Corporation, Carlsbad, CA, USA, cat #A3160502), 1 mM sodium pyruvate (Invitrogen—Life Technologies Corporation, Carlsbad, CA, USA, cat #11360070-100 mM) and 1× non-essential amino-acid supplement, Invitrogen cat #11140050-100x), at 37 °C with 5% CO_2_, until confluent. Cells were plated on ECM arrays (Advanced BioMatrix, Carlsbad, CA, USA; ECM Select^®^ Array Kit Ultra-36 cat #5170) according to the vendor’s instructions; briefly, the array slide was washed three times using 5 mL of PBS, followed with a 5 mL cell-culture medium wash. Then, 250,000 Caco2 cells were suspended in 5 mL of cell-culture media and added to the array slide in the plate provided. Cells were grown for 24 h at 37 °C with 5% CO_2_.

### 2.3. Immunofluorescence Staining and Imaging

Coverslips and ECM array slides were fixed using ice-cold methanol (Thermo Fisher Scientific Inc., NJ, USA, cat #A4524) for 7 min at −20 °C, blocked using blocking reagent (Dako—Agilent Technologies, Inc., Santa Clara, CA, USA, cat#X090930-2) for 1 h at room temperature, and stained with primary antibodies diluted in antibody diluent (Dako - Agilent Technologies, Inc., Santa Clara, CA, USA, cat #S302281-2), overnight at 4 °C. Primary antibodies used in the study were: E-cadherin 4A2C7—Life Technologies Corporation, Carlsbad, CA, USA, cat #33-4000 (1:500 dilution), PLEKHA7—Sigma-Aldrich Inc., St. Louis, MO, USA, cat #HPA038610 (1:100 dilution), DROSHA—Life Technologies Corporation, Carlsbad, CA, USA, cat #MA5-32818 (1:100 dilution) and Cell Signaling Technology Inc., Danvers, MA, USA, cat #D28B1-3364 (1:100 dilution), AGO2—Abcam, Waltham, MA, USA, cat #ab156870 (1:50 dilution). Then, coverslips and ECM array slides were washed three times with PBS (Corning Inc.—Mediatech Inc., Manassas, VA, USA, cat #21040CV) and stained with fluorescently labeled secondary antibodies diluted in antibody diluent for 1 h at room temperature. Secondary antibodies used were Alexa Fluor 488 anti-mouse —Life Technologies Corporation, Carlsbad, CA, USA, cat #A21202 (1:500 dilution) and Alexa Fluor 594 anti-Rabbit—Life Technologies Corporation, Carlsbad, CA, USA, cat #A-11005 (1:500 dilution). Next, coverslips and ECM array slides were washed two times with PBS, co-stained with DAPI (Sigma-Aldrich Inc., St. Louis, MO, USA, cat #D8417, 1: 10,000 dilution) diluted in PBS, and washed once again with PBS before mounting (Polysciences Inc, Aqua Poly/Mount, Warrington, PA, USA, cat #18606). Coverslips and slides were imaged using Leica SP8 confocal microscope (Leica Biosystems, Wetzlar, Germany) under a 63× objective, with an additional 1.5× zoom, Z-stack acquisition intervals at 0.5–1 μm, image resolution at 1024 × 1024 pixels, 600 Hz scan speed, and scan line average at 4. 

### 2.4. Proximity Ligation Assay (PLA)

Confluent cells were fixed either by using ice-cold methanol (Thermo Fisher Scientific Inc., Fair Lawn, NJ, USA, cat #A4524) for 7 min at −20 °C, or by using 4% paraformaldehyde (Electron Microscopy Sciences, Hatfield, PA, USA, cat #157-4) and permeabilized with 0.2% Triton-X (Thermo Fisher Scientific Inc., Fair Lawn, NJ, USA, cat #BP151100) in PBS (Thermo Fisher Scientific Inc., Fair Lawn, NJ, USA, cat #21040CV) at room temperature. Coverslips were blocked using a blocking reagent (Dako—Agilent Technologies, Inc., Santa Clara, CA, USA, cat #X090930-2) for 1 h at room temperature and probed with primary antibodies diluted in antibody diluent (Dako—Agilent Technologies, Inc., Santa Clara, CA, USA, cat #S302281-2) overnight at 4 °C. Primary antibodies used were PLEKHA7—Sigma-Aldrich Inc., St. Louis, MO, USA—HPA038610 (1:100 dilution), p120-catenin-N-terminus—Life Technologies Corporation, Carlsbad, CA, USA, cat #6H11-33-9700 (1:100 dilution), AGO2—ECM Biosciences, Versailles, KY, USA, cat #AP5281 (1:100 dilution), GW182—Santa Cruz Biotechnology Inc., Dallas, TX, USA, cat #TNRC6A-A-8-sc-377006 (1:100 dilution), DROSHA—Cell Signaling Technology Inc., Danvers, MA, USA, cat #D28B1-3364 (1:100 dilution), and DGCR8—Abnova Corporation, Taipei City, Taiwan, cat #H00054487-M01 (1:100 dilution). Antibody combinations used were PLEKHA7 + p120; AGO2 + GW182; and DROSHA + DGCR8. For negative controls, we used coverslips where only one of the two antibodies from each pair was added to the assay. Coverslips were then washed with Duolink^®^ In Situ Wash Buffer-A (Sigma-Aldrich—Duolink^®^- St. Louis, MO, USA, cat #DUO82049) and incubated with a Duolink^®^ Anti-mouse plus PLA probe (Sigma-Aldrich—Duolink^®^, St. Louis, MO, USA, cat #DUO92001) together with a Duolink^®^ Anti-rabbit minus PLA probe (Sigma-Aldrich—Duolink^®^, St. Louis, MO, USA, cat #DUO92005) at 37 °C, for one hour, in a humidified chamber. Coverslips were then washed with Duolink^®^ In Situ Wash Buffer-A (Sigma-Aldrich, cat #DUO82049), incubated with ligation solution (Sigma-Aldrich—Duolink^®^, St. Louis, MO, USA, cat #DUO92013) for 30 min at 37 °C, washed with Duolink^®^ In Situ Wash Buffer-A, and incubated with polymerase solution (Sigma-Aldrich—Duolink^®^, St. Louis, MO, USA, cat #DUO92013) for 100 min at 37 °C. Coverslips were then washed with Duolink^®^ In Situ Wash Buffer-B (Sigma-Aldrich—Duolink^®^, St. Louis, MO, USA, cat #DUO82049), then with Duolink^®^ In Situ Wash Buffer-A and counterstained with Alexa flour 488 conjugated E-cadherin antibody (Cell Signaling Technology Inc., Danvers, MA, USA, cat #3199S) for 1 h at room temperature. Coverslips were eventually washed with Duolink^®^ In Situ Wash Buffer-B and with Duolink^®^ In Situ Wash Buffer-A before being mounted with Duolink^®^ In Situ Mounting Medium with DAPI (Sigma-Aldrich—Duolink^®^, St. Louis, MO, USA, cat #DUO82040). Coverslips were imaged using a Leica SP8 (Leica Biosystems, Wetzlar, Germany) confocal microscope under 63× objective with an additional 1.5× zoom, with Z-stack acquisition intervals at 0.5–1 μm, image resolution at 1024 × 1024 pixels, 600 Hz scan speed, and scan line average at 4.

### 2.5. Image Analysis and Quantifications

Immunofluorescence images were analyzed using Fiji. The goal of these analyses was to evaluate changes in the junctional localization of the markers of interest. When mis-localized from the junctions, PLEKHA7 and AGO2 can also localize to the cytoplasm, whereas DROSHA can localize to the cytoplasm and the nucleus. To detect these changes in localization and to control for any experimental variability between different fields or experiments, we calculated the junctional/cytoplasmic ratios of the fluorescence intensities of PLEKHA7, AGO2, or the junctional/(cytoplasmic + nuclear) fluorescence intensities for DROSHA. To do this, we used maximum (max) projections of Z-stacks and calculated the fluorescence intensities of 6 µM—long straight lines either along a triplicate of junctions (Figure 1, lines 1–3) or within a triplicate of adjacent cytoplasmic regions (Figure 1, lines 4–6). Ratios were then calculated by dividing the total junctional (1 + 2 + 3) by the total cytoplasmic fluorescence intensities (4 + 5 + 6). We measured three cells per field, for three fields, totaling 27 cells per condition. Since DROSHA also exhibits nuclear localization, we measured junctional (Figure 1, lines 7–9), cytoplasmic (Figure 1, lines 10–12), as well as nuclear fluorescence intensities (Figure 1, lines 13–15), and calculated the ratio of junctional fluorescence to the sum of cytoplasmic + nuclear fluorescence. In all cases, we measured three cells per field, for three fields, totaling 27 cells per condition.

To calculate tricellular junction vs. bicellular junction fluorescence intensity ratios, we divided the fluorescence intensities of boxes covering either the whole area of a tricellular junction (Figure 1, box 16) or the same area of a bicellular junction (Figure 1, box 17). We repeated this for three tricellular junctions and three bicellular junctions per cell, for three cells per field, and for three fields, for a total of 27 measurements per condition. 

To calculate the co-localization of PLEKHA7, AGO2, or DROSHA with E-cadherin, we drew boxes of 28 µm^2^ along junctions (Figure 1, box 17) and measured co-localization using the JaCoP function in Fiji to obtain the Pearson’s coefficient from a minimum of 12 to a maximum of 30 cells from three different fields, per condition.

For Proximity Ligation (PLA) quantifications, we counted the total number of positive signals along the junctions (Figure 1, indicated by #18), the total number of cytoplasmic signals (Figure 1, indicated by #19), as well as the total number of nuclear signals for the DROSHA-DGCR8 PLA (Figure 1, indicated by #20) in the same cells, using the “analyze particle” function in Fiji. Then, we calculated the ratio of junctional/cytoplasmic or junctional/(cytoplasmic + nuclear) (for DROSHA-DGCR8) PLA signals to assess the junctional complex formation of PLEKHA7-p120 and RNAi machinery complexes. This process was repeated for a minimum of 9 to a maximum of 12 cells across three different fields per condition. In addition, we calculated the ratio of the total number of PLA signals/cell numbers to measure the overall differences in complex formation between different conditions.

To evaluate junction linearity, we traced the actual junction to measure its length using Fiji (Figure 1, line 21) and then divided this length by the length of the straight linear junction (Figure 1, line 22) that can form between the two ends of the same junction. We repeated this measurement for three junctions of a cell, for three cells per field, and for three fields, totaling 27 cells per condition. 

To validate the significance of differences in the above-mentioned measurements, we used a student *t*-test and compared the results from fibronectin, collagen I, and collagen IV to that of laminin. To analyze the results from the ECM combination experiments we utilized one-way ANOVA, where we used the Shapiro-Wilk test for the normality test and Bonferroni correction for pair-wise comparisons. In all the above measurements, we set a scale to keep the pixel/micron ratio constant. All experiments were performed in three biological replicates or two regarding PLA.

### 2.6. Immunoblotting

Whole-cell extracts were obtained using the RIPA buffer (50 mM Tris, pH 7.4—Bio-Rad Laboratories Inc, Hercules, CA, USA, cat #1610719; 150 mM NaCl—Sigma-Aldrich Inc., St. Louis, MO, USA, cat #S9888- 10K; 1% NP-40-Fisher Scientific, cat #507517565; 0.5% deoxycholic acid, Sigma-Aldrich Inc., St. Louis, MO, USA, cat #D6750-100G; and 0.1% SDS, Thermo Fisher Scientific Inc., Ottawa, Canada, cat #BP166-500), supplemented with protease (Thermo Fisher Scientific Inc, Carlsbad, CA, USA, cat #50550432) and phosphatase (Pierce Biotechnology Inc., Rockford, IL, USA, cat #P178420) inhibitors. Lysates were homogenized by passing through a 29-G needle and cleared by full-speed centrifugation for 5 min. Protein quantification was performed using a Pierce BCA Protein Assay (Pierce Biotechnology Inc., Rockford, IL, USA, cat #I23227). Protein extracts were mixed with Laemmli sample buffer at 2x final, separated by SDS-PAGE using 4–20% TGX gels (Bio-Rad, Hercules, CA, USA, cat #4568094), and transferred to 0.2 μm nitrocellulose membranes (Bio-Rad Laboratories Inc, Hercules, CA, USA, cat #1704158) with the Bio-Rad^®^ Trans-Blot Turbo Transfer System. Membranes were blocked and blotted in 3% milk according to standard protocols. Antibodies used were E-cadherin—4A2C7 Invitrogen-Life, cat #33-4000 (1:2000 dilution), PLEKHA7—Sigma Aldrich, cat #HPA038610 (1:1000 dilution), DROSHA—Invitrogen-Life, cat #MA5-32818 (1:1000 dilution), and Cell Signaling, cat #D28B1-3364 (1:1000 dilution), AGO2—Abcam, cat #ab156870 (1:1000 dilution), GAPDH—Cell Signaling Technology Inc., Danvers, MA, USA, cat #14C10-2118 (1:2000 dilution), and Actin—Cell Signaling Technology Inc., Danvers, MA, USA, cat #4967L (1:2000 dilution). Signals were detected by luminescence using Pierce ECL (Bio-Rad, Hercules, CA, USA, Pierce—cat #32209) using a Bio-Rad^®^ ChemiDoc Imaging System. Quantifications were performed using Fiji.

## 3. Results

### 3.1. Fibronectin and Collagen I Negatively Impact Junctional Localization of PLEKHA7 and the Key RNAi Component AGO2

To examine the effects of ECM components on the subcellular localization of PLEKHA7 and of core components of the RNAi machinery, we analyzed immunofluorescence images of Caco2 colon epithelial cells grown separately on laminin, fibronectin, collagen I, and collagen IV, which are key ECM substrates found in the colonic mucosa [1,2,3,4]. Since the junctional localization of PLEKHA7 is critical for recruiting RNAi components to the AJs [62,63], we first calculated the junctional/cytoplasmic ratio of PLEKHA7. PLEKHA7 primarily localizes at mature AJs that form at the apical areas of cell–cell contact in epithelial monolayers. However, here, we used the max projections of our confocal microscopy - acquired image stacks for these calculations, so that we capture the whole of the potential cytoplasmic PLEKHA7, since this could be distributed throughout the cell, if mis-localized (Figure 2). We followed the same strategy for the remaining markers we examined throughout the manuscript. We then calculated and used the junctional/cytoplasmic ratios for our comparisons, instead of absolute fluorescent signal intensities, to (a) detect shifts in the subcellular distribution in each experiment and (b) to control for any fluctuations in fluorescence intensities between experiments. This approach revealed that cells exhibited the highest junctional/cytoplasmic ratio of PLEKHA7 when plated either on laminin or collagen IV, whereas that ratio was significantly lower in cells plated on fibronectin and collagen I, when compared to laminin (Figure 2A,B). In agreement, co-localization of PLEKHA7 with E-cadherin was also significantly lower in cells plated on fibronectin and collagen I, compared to laminin (Figure 2C), although the average junctional/cytoplasmic fluorescence ratio of E-cadherin was not significantly altered in these conditions (Figure 2D). Overall, total protein levels of PLEKHA7 or E-cadherin were not affected when cells were plated on different ECM substrates (Figure 2E), demonstrating that the observed differences strictly reflect changes in subcellular distribution. Therefore, certain ECM substrates, such as fibronectin and collagen I, negatively impact localization of PLEKHA7 to the junctions, without altering the core structure of AJs, as indicated by the E-cadherin findings.

Following the same approach, we examined whether subcellular localization of the key RNAi machinery component AGO2 is also affected by different ECM substrates. We have previously shown that junctional PLEKHA7 is required for the recruitment and function of AGO2 to the AJs [63,65]. Indeed, following the same pattern with PLEKHA7, junctional distribution of AGO2 was significantly decreased in cells plated on fibronectin and collagen I, compared to cells plated on laminin and collagen IV (Figure 3A,B). This is also indicated by lower levels of AGO2 co-localization with E-cadherin (Figure 3C), even though the protein levels of AGO2 were not significantly changed in any of these conditions (Figure 3D). Therefore, junctional localization of both PLEKHA7 and AGO2, but not their overall levels, is similarly sensitive to cells plated in different ECM substrates, and negatively impacted particularly by the pro-fibrotic, pro-tumorigenic fibronectin and collagen I. 

### 3.2. ECM Substrates Differently Affect Subcellular Localization of DROSHA, as Well as Its Spatial Distribution across Areas of Cell–Cell Contact

We then sought to examine the effects of different ECM substrates on the subcellular localization of DROSHA, following the same approach as with PLEKHA7 and AGO2. In the case of DROSHA, we calculated the ratio of junctional versus the cytoplasmic-plus-nuclear fluorescent signal, since DROSHA also exhibits nuclear localization [77,78,79] (see also Figure 1 and Section 2.5). Similarly to what we observed with PLEKHA7 and AGO2, junctional localization of DROSHA is decreased when cells are plated on fibronectin, compared to laminin; however, this decrease is less profound when cells are plated on collagen I (Figure 4A,B). A similar pattern was found when calculating the co-localization of DROSHA with E-cadherin (Figure 4C). Overall protein levels of DROSHA remained the same, when plated on either one of these ECM substrates (Figure 4D), again demonstrating that any differences observed by immunofluorescence reflect changes in the subcellular distribution of the protein.

In addition to showing distinct subcellular localization patterns, we observed accumulation of DROSHA at apical tricellular junctions in collagen I—plated cells compared to apical bicellular linear junctions of these cells (Figure 4A). Tricellular junctions are under higher contractile actomyosin tension than bicellular junctions and are essential for regulating overall mechanical signals, cell polarity, and epithelial barrier function [80,81,82]. Calculating the fluorescence ratio between tricellular and bicellular junctions shows that cells plated on collagen I seem to accumulate more DROSHA at tricellular junctions compared to cells plated on laminin, fibronectin, and collagen IV (Figure 4E). Notably, we also observed DROSHA localization at basal areas of cell–cell contact of cells grown on collagen I and collagen IV (Figure 5), revealing a broader distribution pattern of DROSHA across apicobasal areas of cell–cell contact when collagens were used as the ECM substrate. Together, these results show that DROSHA not only exhibits distinct distribution across different subcellular compartments but also distinct distribution across areas of cell–cell contact in response to different ECM substrates, differently from what we observed with PLEKHA7 and AGO2.

### 3.3. Different ECM Substrates Affect RNAi Complex Formation

Since the four ECM substrates we used to culture Caco2 cells impact localization of PLEKHA7 and RNAi components to the junctions, we then asked whether they also impact their functional interactions at the junctions. For example, PLEKHA7′s interaction with p120 is critical for its recruitment to the adherens junctions [83]. The interacting partner of AGO2 that is essential for its function within RISC is GW182, and the critical functional partner of DROSHA is DGCR8, with which forms the microprocessor complex [61]. Therefore, we examined the effects of the four different ECM substrates we used in this study (laminin, fibronectin, collagen I, and collagen IV) on the formation of these three interacting pairs, namely PLEKHA7-p120, AGO2-GW182, and DROSHA-DGCR8. To do this, we employed a proximity ligation assay (PLA). This assay provides advantages over a co-immunoprecipitation assay in that it: a) allows visualization within the cell, enabling assessment of subcellular localization; and b) is less susceptible to non-specific opportunistic interactions that may occur during co-immunoprecipitation, which takes place after cell lysis and conditions like plating on ECM substrates, are removed. To evaluate complex formation of these pairs at the junctions, we performed PLA and then measured either the junctional/cytoplasmic (for PLEKHA-p120 and AGO2-GW182) or the junctional/(cytoplasmic + nuclear) (for DROSHA-DGCR8) ratios of PLA signal counts (see also Figure 1 and Section 2.5). In all cases, the junctional localization of PLA signals was assessed by co-immunofluorescence with E-cadherin, which was performed immediately upon completion of PLA (see Section 2.4). This approach revealed robust interactions between PLEKHA7-p120, AGO2-GW182, and DROSHA-DGCR8 at the junctions of cells plated on laminin (Figure 6A). Fibronectin and collagen I disrupted all three interactions at the junctions, compared to laminin (Figure 6A,B), which is in agreement with the disruption of the overall junctional localization of these components by these ECM substrates (Figure 2, Figure 3 and Figure 4). However, PLA revealed that collagen IV also negatively impacts all three interactions at the junctions (Figure 6A,B), although it didn’t seem to affect their overall junctional localization (Figure 2, Figure 3 and Figure 4). In fact, we noticed that both collagens, as well as fibronectin, seem to negatively affect these interactions not only at the junctions but throughout the cell. Indeed, total PLA signal counts per cell (Figure 6A,C) confirmed this observation, with the only exception of the overall total PLEKHA7-p120 interactions of cells plated on fibronectin, which were unaffected (Figure 6A,C). In that later case, fibronectin still decreases the junctional PLEKHA7-p120 interactions (Figure 6A,B). Together, these results show that fibronectin and collagens I and IV not only negatively affect complex formation of PLEKHA7-p120, AGO2-GW182, and DROSHA-DGCR8 at the junctions, but also their overall interactions within cells.

### 3.4. Different ECM Substrates Affect Junction Linearity

Upon close examination of confocal images of cells plated on the four different ECM substrates that we used, we noticed that apical junctions were more wriggled in the cells plated on fibronectin and collagen I (Figure 2A). Indeed, by calculating the ratio of the actual junctional distance between two tricellular junctions vs. their shortest possible linear length (see also Methods, Section 2.5 and Figure 1), we confirmed that cells plated on collagen I and fibronectin had significantly less linear junctions compared to the cells plated on laminin and collagen IV (Figure 7). This observation implies that fibronectin and collagen I might affect apical junctional tension. Indeed, apical junction linearity is the result of tensile AJs, whereas wriggled cell–cell contacts indicate a disruption in AJ tension [84]. Tension at AJs is established through interaction of cadherin-catenin junctions with the actomyosin cytoskeleton, which is critical for maintaining junction integrity and cell shape [85,86,87,88]. Thus, disruption in linearity by fibronectin and collagen I suggests defects in junction formation and possibly compromised epithelial integrity, in addition to negatively affecting the localization of PLEKHA7 and AGO2 to the AJs. 

### 3.5. ECM Substrate Combinations also Differentially Affect Localization of PLEKHA7 and RNAi Components to AJs

We have found that the junctional localization of PLEKHA7, AGO2, and DROSHA are differently impacted when plating cells in different ECM substrates. Since the relative composition of these ECM components is altered during fibrotic conditions or cancer, we also examined the junctional localization of PLEKHA7 and RNAi components upon plating epithelial cells to combinations of laminin, fibronectin, collagen I, and collagen IV. To do this, we plated Caco2 cells on a commercially available array with different combinations and concentrations of these ECM substrates (Advanced BioMatrix; see Methods, Section 2.1) and stained cells by immunofluorescence for PLEKHA7, AGO2, and DROSHA, as above. By performing similar quantifications of immunofluorescence images, we found that the average junctional/cytoplasmic fluorescence ratio of both PLEKHA7 and AGO2 is significantly less in cells grown on combinations of all four substrates (Figure 8A,B and Appendix A) compared to cells plated on laminin only. In addition, the average junctional/cytoplasmic fluorescence ratio of AGO2 is significantly less in cells grown on combinations of fibronectin + laminin, fibronectin + laminin + collagen I, and fibronectin + laminin + collagen IV compared to cells plated on laminin only (Figure 8A,B and Appendix A). Together, combinations that negatively affected the junctional localization of PLEKHA7 and AGO2 have fibronectin in common. Notably, the average junctional/cytoplasmic fluorescence ratio of PLEKHA7 and AGO2 is also significantly less in cells grown on collagen IV (Figure 8A,B and Appendix A). This may be due to the higher concentration of collagen IV used in this array. Indeed, when plating cells only on collagen IV, but at this higher concentration (250 µg/mL), PLEKHA7 and AGO2 are significantly less junctional compared to laminin - only plated cells of the same concentration (Figure 8A,B). However, this localization was not affected when cells were plated on the collagen IV concentrations we used above in our individually coated coverslips (45 µg/mL; Figure 2A,B and Figure 3A,B). Localization of DROSHA exhibited similar trends of fluorescence ratios with PLEKHA7 and AGO2 in cells plated on the same substrate combinations, but these changes did not reach significance, when compared to those of cells grown on laminin (Figure 8C and Appendix A).

Taken together, our results show that ECM substrates can impact localization and interactions of PLEKHA7 and RNAi components at AJs, with fibronectin, collagen I, and high concentrations of collagen IV being overall negative regulators. They also show that DROSHA exhibits patterns of localization both intracellularly and across areas of cell–cell contact that are distinct from PLEKHA7 and AGO2.

## 4. Discussion

Our previous studies have shown that PLEKHA7 recruits core RNAi machinery components, including AGO2 and DROSHA, to mature apical AJs, thereby regulating the levels of a set of miRNAs and mRNAs [62,63]. Depletion of PLEKHA7 causes mislocalization of AGO2 and DROSHA from the junctions, downregulation of homeostatic miRNAs, increased expression of oncogenes, and anchorage-independent growth of Caco2 cells [62,63]. Moreover, we have reported that the junctional localization of PLEKHA7 and RNAi components is broadly disrupted in colon tumors [65]. Together, these results suggest a localized, tumor-suppressing mechanism at epithelial AJs. However, the overall modes of regulation of this mechanism are still not defined.

Our current data show that ECM components can impact RNAi localization and complex formation at the junctions, adding one more element in the interplay between ECM and AJs. More specifically, we show that fibronectin and collagen I negatively impact junctional localization of PLEKHA7 and AGO2 when compared to laminin and relatively moderate concentrations of collagen IV (Figure 2, Figure 3 and Figure 4). Since laminin and collagen IV are components of the basement membrane, epithelial cells primarily attach to these substrates under homeostatic conditions. Proteins in the basement membrane are evolutionarily conserved in mammals [89,90,91]. Therefore, cells might be sensing a “normal”, homeostatic stromal composition when plated on laminin and collagen IV, consequently maintaining junction integrity. However, cells can be exposed to fibronectin and collagen I, e.g., after an insult to the epithelial barrier, resulting in the mislocalization of PLEKHA7 and RNAi from cell–cell junctions. This could be beneficial at first since cells would rather activate pro-proliferative and pro-growth regulators in that case to close the wound, such as those suppressed by the junctional RNAi machinery, including SNAI1, JUN, and MYC [62,63]. However, previous studies have also demonstrated that increased fibronectin deposition is a biomarker for cancer and a common feature of advanced tumors [28,29], whereas collagen I deposition is also increased in tumors and fibrotic tissues [26]. Since our current findings indicate that fibronectin and collagen I can trigger mislocalization of PLEKHA7 and AGO2 from the junctions, it is likely that these ECM substrates may contribute to pro-tumorigenic behavior of epithelial cells, at least in part, through dysregulation of junctional RNAi complexes and upregulation of these same pro-proliferative markers, which are also well-established oncogenes. 

Interestingly, PLA results show that collagen IV, together with fibronectin and collagen I, also negatively impacts junctional complex formation of PLEKHA7-p120, as well as of AGO2-GW182 and DROSHA-DGCR8, which are essential functional interactions for the RNAi machinery (Figure 6). In addition, the ECM array results show that when the concentration of collagen IV is higher (125–250 µg/mL; Figure 8A,B and Appendix A), compared to our individually coated coverslips (Figure 2 and Figure 3; 45 µg/mL), can also negatively impact junctional localization of PLEKHA7 and AGO2, even in combination with laminin (Figure 8A,B). Indeed, increased collagen IV levels have been observed and shown to contribute to liver metastasis and fibrosis [72,73,74,75]. In addition, targeting collagen IV reduced tumor volume in mouse xenografts [92]. Further, abnormal collagen IV can induce EMT in healthy epithelial cells [93]. Therefore, it seems that high levels of collagen IV may also contribute to pro-tumorigenic transformation through dysregulation of the junctional RNAi machinery.

Since there is extensive crosstalk between the ECM and the AJs via the actin cytoskeleton [36,37,38], it is possible that the effects of fibronectin and collagens on the junctional localization of RNAi components are mediated by the actin cytoskeleton. We have recently shown that loss of cortical actomyosin tension at AJs disrupts AGO2 recruitment to areas of cell–cell contact [94]. Indeed, the junctional localization of AGO2 to the junctions is sensitive to fibronectin and collagens (Figure 3 and Figure 6). However, fibronectin and collagen I not only negatively impact junctional localization of DROSHA and its interaction with DGCR8, but also alter its spatial distribution along areas of cell–cell contact. In particular, DROSHA exhibits strong localization at tricellular junctions of Caco2 cells when grown on collagen I (Figure 4), whereas it translocates to basal cell–cell junctions of Caco2 cells when plated on collagen I and IV (Figure 5). Furthermore, DROSHA did not exhibit significant changes in localization plated in combinations of these substrates, contrary to PLEKHA7 and AGO2 (Figure 8C). Together, these results imply a mechanism regulating subcellular localization of DROSHA that is distinct from that regulating localization of PLEKHA7 and AGO2. For example, recent research has shown that canoe, a cytoskeleton interacting protein and the *Drosophila* homolog of afadin associated with PLEKHA7 [95], is recruited to the tricellular junctions during epithelial remodeling in the *Drosophila* embryo in response to mechanical tension, thereby stabilizing tricellular junctions [96]. Therefore, it is likely that DROSHA’s tricellular junction localization is in response to the altered distribution of mechanical tension of the epithelial layer generated by collagen I and fibronectin. These results suggest that different ECM substrates may be differently affecting apical junctional tension, thus differently affecting the localization of RNAi components to areas of cell–cell contact. In addition to these findings, our data also show that fibronectin, collagen I, and collagen IV negatively impact AGO2-GW182 and DROSHA-DGCR8 interactions not only at the junctions but also throughout the cell, revealing an additional layer of regulation (Figure 6). These are essential interactions that are required for miRNA processing (DROSHA-DGCR8) and miRNA-mediated mRNA silencing (AGO2-GW182) in the cells [61]. Together, these observations beg the question of the distinct mechanisms that fine-tune subcellular localization and function of the core RNAi complexes, such as the microprocessor (DROSHA-DGCR8) and RISC (AGO2-GW182), which warrants further investigation.

In summary, this study demonstrates that differences in the ECM composition, which may occur during diseases such as fibrosis or cancer, can affect junctional localization and potentially function of the RNAi machinery. Our study has limitations in that a) it did not account for any possible post-translational modifications of the commercially available ECM substrates used, which are of different origin; b) we used a pre-determined set of combinations of ECM substrates in Figure 8, although variability in these combinations is much broader in tissues; and c) we did not account for the three-dimensional organization and alignment of collagen I fibers, which has been shown to promote pro-tumorigenic behavior [54,60,97]. Still, our findings provide a basis to investigate the effects of ECM composition and organization on RNAi complex formation and function, which in turn can influence cell behavior post-transcriptionally, even without any direct genetic alterations. Similar crosstalk has been recently reported in endothelial cells and fibroblasts, where ECM stiffness can alter miRNA function [98]. Such crosstalk between the ECM, the RNAi machinery, and miRNAs can play a mechanistic role as a trigger in the progression from fibrotic tissue to neoplasia or from tumorigenesis to tumor progression and cell migration by predisposing cells to pro-tumorigenic behavior, which can then further progress with the additional occurrence of genetic mutations. Indeed, several studies have shown that the stroma can promote such behaviors of tumor cells, also in an E-cadherin-p120 catenin—dependent manner [99]. We have also shown the existence of distinct E-cadherin-p120 catenin complexes, a tumor-suppressing that includes RNAi complexes, and a tumor-promoting that is absent of RNAi [62]. Therefore, in this context, it would be interesting to examine whether ECM alterations promote pro-tumorigenic behaviors by tilting the balance of cadherin complexes from a tumor-suppressing to a tumor-promoting one, which will further elucidate the role of cell–cell adhesion complexes in the progression of epithelial diseases.

## Figures and Tables

**Figure 1 cells-11-03740-f001:**
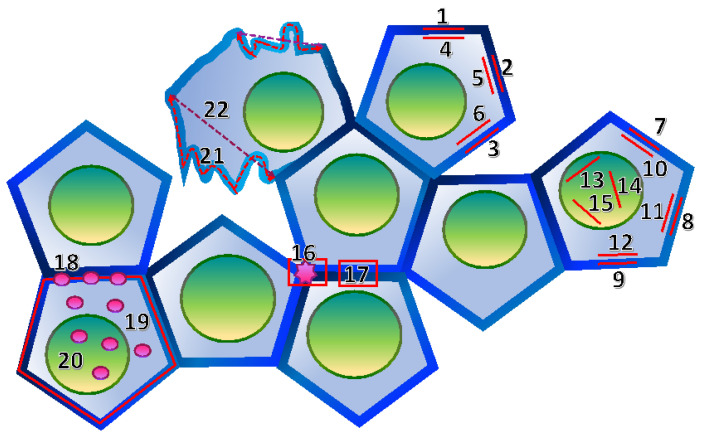
Schematic outlining the quantification strategy for evaluating junctional/cytoplasmic localization ratios of PLEKHA7, AGO2 (lines 1–6); junctional/(cytoplasmic + nuclear) localization ratio of DROSHA (lines 7–15); tricellular vs. bicellular fluorescence intensities (boxes 16,17); proximity ligation (PLA) junctional (18) vs. cytoplasmic (19) and nuclear (20) signals; and linear vs. wriggled junctional lengths (lines 21–22). Star indicates a tri-cellular junction. See text, Section 2.5, for details.

**Figure 2 cells-11-03740-f002:**
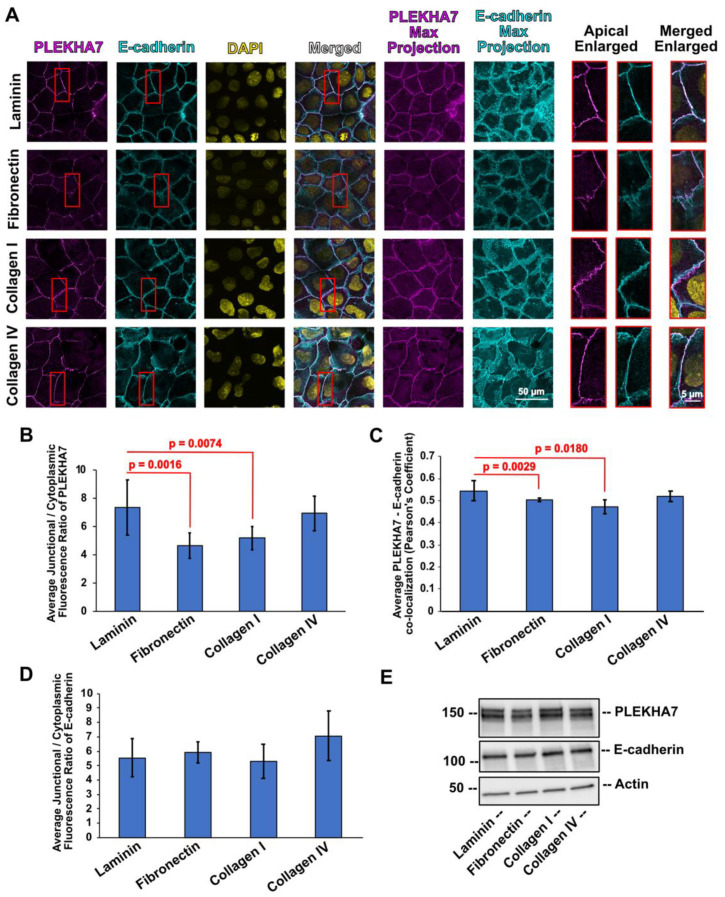
(**A**) Representative images of Caco2 colon epithelial cells plated on laminin, fibronectin, collagen I, and collagen IV, stained by immunofluorescence for PLEKHA7 (magenta) and E-cadherin (cyan). Nuclei were stained using DAPI (yellow). (**B**) Average junctional/cytoplasmic fluorescence intensity ratios of PLEKHA7 from cells plated on different ECM substrates. (**C**) Average co-localization of PLEKHA7 with E-cadherin fluorescence from cells plated on different ECM substrates. (**D**) Average junctional/cytoplasmic fluorescence intensity ratio of E-cadherin from cells plated on different ECM substrates. (**E**) Western blot from Caco2 cells plated on different ECM substrates blotted for PLEKHA7 and E-cadherin; Actin is the loading control. Data were analyzed using a student *t*-test, where *p* < 0.05 and *n* = 27 or *n* = 12 ≤ *n* ≤ 30 for (**C**) (see also Methods, Section 2.5). Selected insets indicated in red boxes are shown enlarged on the right.

**Figure 3 cells-11-03740-f003:**
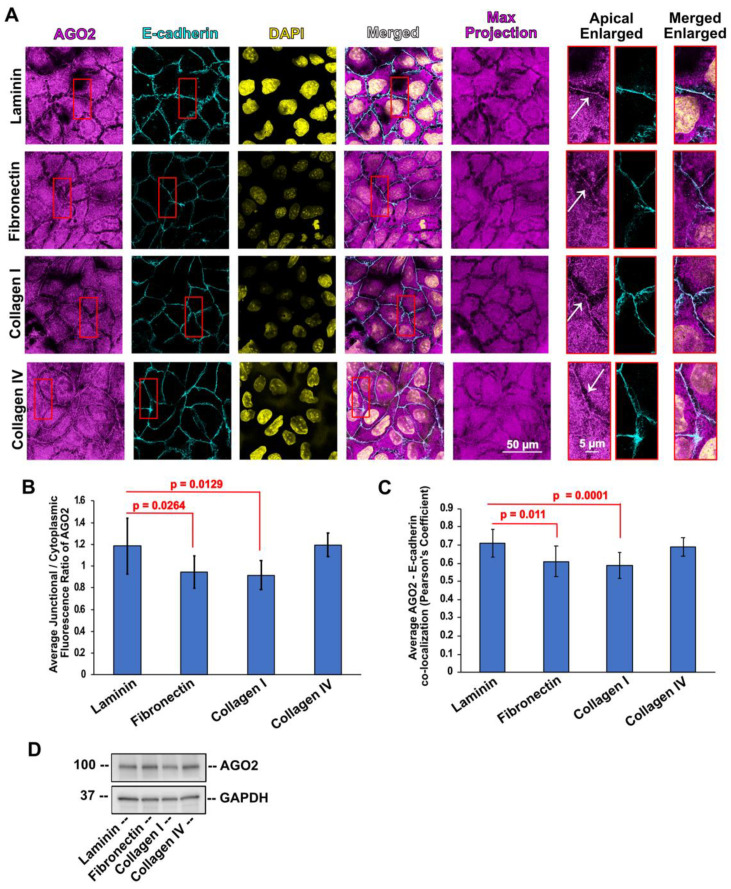
(**A**) Representative images of Caco2 colon epithelial cells plated on laminin, fibronectin, collagen I, and collagen IV, stained by immunofluorescence for AGO2 (magenta) and E-cadherin (cyan). Nuclei were stained using DAPI (yellow). (**B**) Average junctional/cytoplasmic fluorescence intensity ratios of AGO2 from cells plated on different ECM substrates. (**C**) Average co-localization of AGO2 with E-cadherin fluorescence from cells plated on different ECM substrates. (**D**) Western blot from Caco2 cells plated on different ECM substrates blotted for AGO2; Actin is the loading control. Data were analyzed using a student *t*-test, where *p* < 0.05 and *n* = 27 or *n* = 12 ≤ *n* ≤ 30 for (**C**) (see also Methods, Section 2.5). White arrows indicate apical junctions. Selected insets indicated in red boxes are shown enlarged on the right.

**Figure 4 cells-11-03740-f004:**
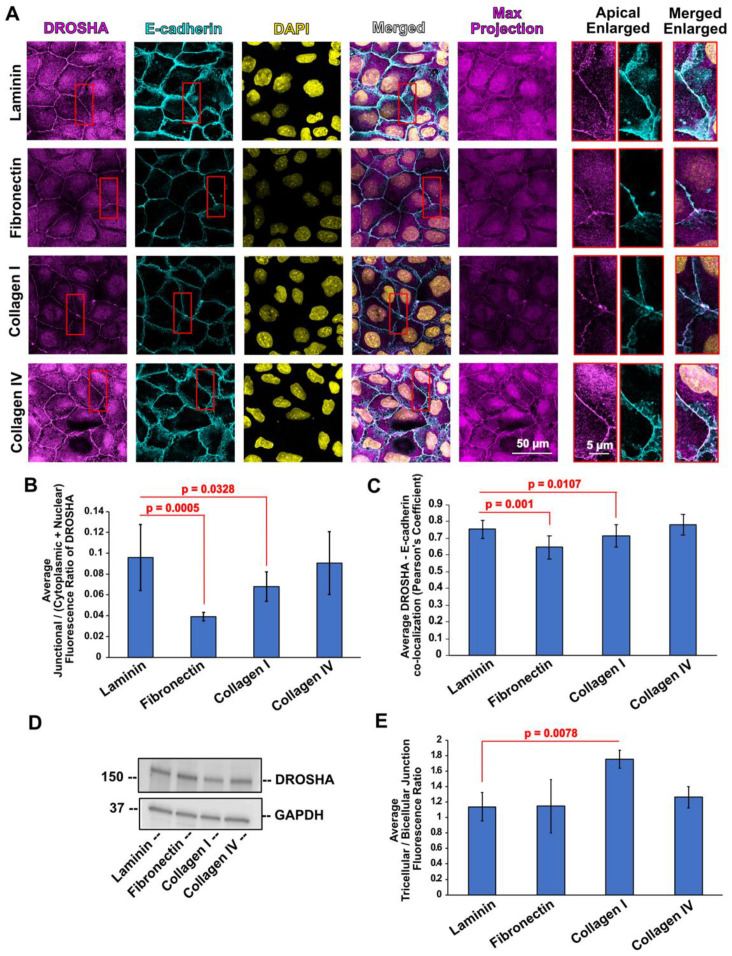
(**A**) Representative images of Caco2 colon epithelial cells plated on laminin, fibronectin, collagen I, and collagen IV stained by immunofluorescence for DROSHA (magenta), and E-cadherin (cyan). Nuclei were stained using DAPI (yellow). (**B**) Average junctional/(cytoplasmic + nuclear) fluorescence intensity ratios of DROSHA from cells plated on different ECM substrates. (**C**) Average co-localization of DROSHA with E-cadherin fluorescence from cells plated on different ECM substrates. (**D**) Western blot of Caco2 cells plated on different ECM substrates blotted for DROSHA; Actin is the loading control. (**E**) Average tricellular/bicellular fluorescence intensity ratios of DROSHA from Caco2 cells plated on different ECM substrates. Data were analyzed using a student *t*-test, where *p* < 0.05 and *n* = 27 or *n* = 12 ≤ *n* ≤ 30 for (**C**) (see also Methods, Section 2.5). Selected insets indicated in red boxes are shown enlarged on the right.

**Figure 5 cells-11-03740-f005:**
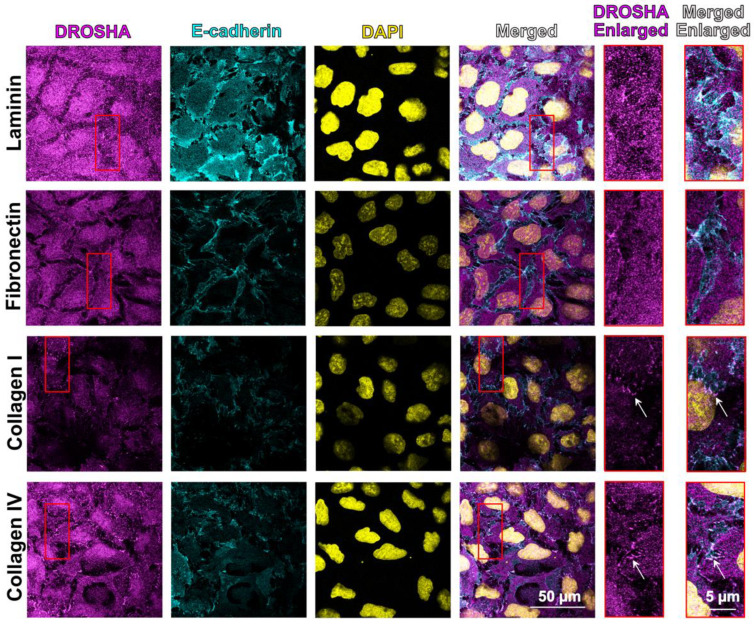
Representative images of basal areas of Caco2 colon epithelial cells plated on laminin, fibronectin, collagen I, and collagen IV stained by immunofluorescence for DROSHA (magenta) and E-cadherin (cyan). Nuclei were stained using DAPI (yellow). Arrows indicate basal areas of cell–cell contact. Selected insets indicated in red boxes are shown enlarged on the right.

**Figure 6 cells-11-03740-f006:**
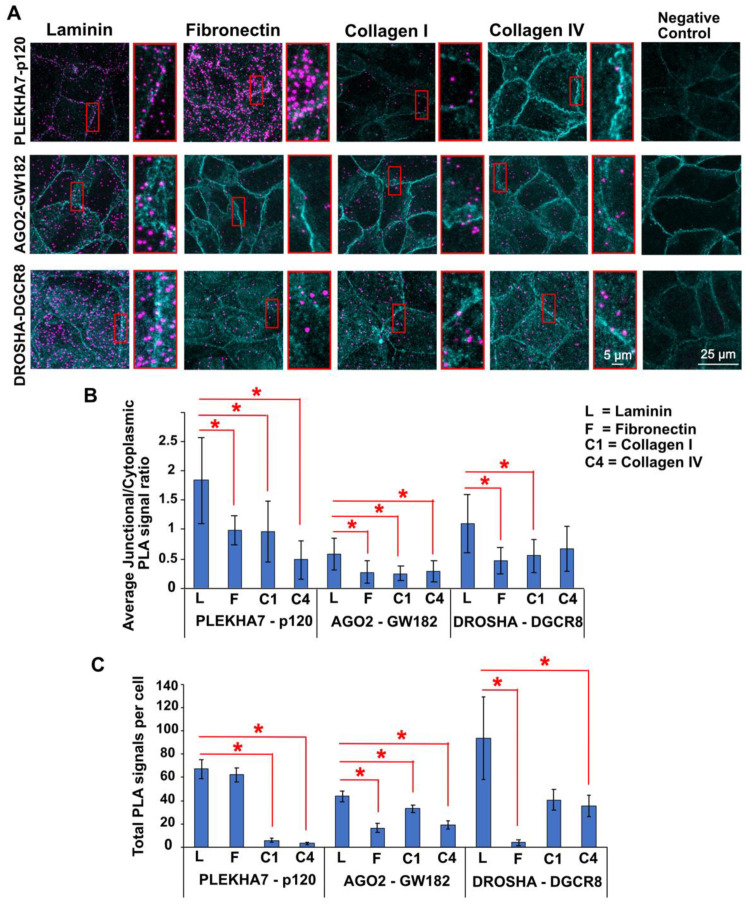
(**A**) Representative images of proximity ligation assay (PLA; magenta) for PLEKHA7-p120, AGO2-GW182, and DROSHA-DGCR8 of Caco2 colon epithelial cells plated on laminin, fibronectin, collagen I, and collagen IV, co-stained by immunofluorescence with E-cadherin (cyan). (**B**) Average junctional/cytoplasmic (for PLEKHA7-p120, AGO2-GW182) or junctional/(cytoplasmic + nucleus) (DROSHA-DGCR8) ratios of PLA signals. (**C**) Total PLA signals per cell count. The data were analyzed using a student *t*-test, where *p* < 0.05 and 9 ≤ *n* ≤ 12. (see also Methods, Section 2.5). Statistical significance of *p* < 0.05 is indicated by asterisks (*). Selected insets indicated in red boxes are shown enlarged on the right of each image column.

**Figure 7 cells-11-03740-f007:**
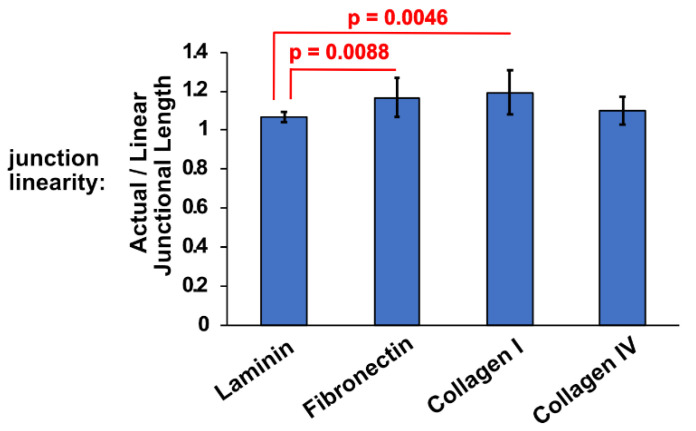
Average actual (wriggled)/linear junctional length (measured in µm) ratio of cells plated on different ECM substrates (for representative images, see Figure 1A). The data were analyzed using a student *t*-test, where *p* < 0.05 and *n* = 27.

**Figure 8 cells-11-03740-f008:**
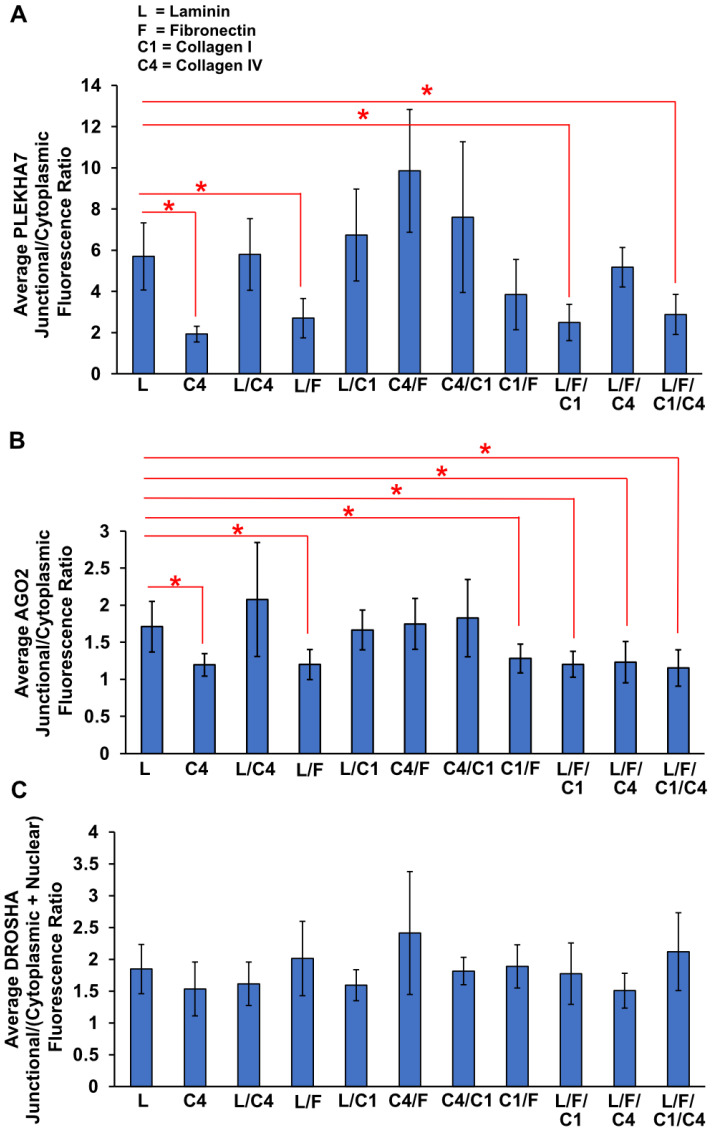
Average junctional/cytoplasmic fluorescence intensity ratios of PLEKHA7 (**A**) and AGO2 (**B**) in Caco2 colon epithelial cells plated on different combinations of ECM substrates, compared to that of laminin. (**C**) Average junctional/(nuclear + cytoplasmic) fluorescence intensity ratio of DROSHA in Caco2 colon epithelial cells plated on different combinations of ECM proteins. Data were analyzed using one-way ANOVA with the Shapiro–Wilk test for the normality test and Bonferroni correction for pair-wise comparison, where *p* < 0.05 and 9 ≤ *n* ≤ 15 (see also Methods, Section 2.5). Statistical significance of *p* < 0.05 is indicated by asterisks (*).

**Table 1 cells-11-03740-t001:** Calculations for different ECM protein solutions.

Substrate	Recommended Concentration (µg/cm^2^)	Used Final Concentration (µg/cm^2^)	Diluted Solution
Fibronectin (Sigma-Aldrich Inc., St. Louis, MO, USA, cat #F1141-1MG)	1–5	2.5	PBS (Corning Inc.—Mediatech Inc., Manassas, VA, USA, cat #21040CV)
Laminin (Sigma-Aldrich Inc., cat #L2020)	1–2	1.5	HBSS (Life Technologies Corporation, Carlsbad, CA, USA, cat #14170-120)
Collagen IV (Advanced BioMatrix, Carlsbad, CA, USA, cat #5022-5MG)	10–100	10	0.25% Acetic acid (Sigma-Aldrich Inc., cat #695092-500 mL)
Collagen I (Sigma-Aldrich Inc., cat #122-20)	5	5	Used the original solution according to the vendor’s recommendation.

## Data Availability

All data are available in this manuscript.

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
