# Peer review of "ECM Substrates Impact RNAi Localization at Adherens Junctions of Colon Epithelial Cells"

_cells, 2022, doi:10.3390/cells11233740_

Round 1
Reviewer 1 Report
1. Could the authors perform the Co-IP experiments to show the difference in protein localization at the adherens junctions on the different substrates?
2. Three figures of Western blot results have the same actin bands for loading control. There is no information about the number of repetitions of this experiment. Was it just one?
3. Original images of western blots:
left top - is it one membrane or several, or bands transferred from 3 gels to one membrane? it is not clear, especially because of the shifted bands.
left bottom - the same question, why additional shifted bands are there, what is it?
Could the authors add some descriptions here to explain it?
Also, it should be obvious that loading control is stained on the same membrane, as the other proteins, but here actin is on the separated membrane.
Could the authors repeat the westerns with the same lysates but with staining actin on the same membrane as proteins of interest?
4. Could the authors add an illustration of how they created the selections for measurements? It is not clear from the description.
5. Figures 2a and 3a: It would be nice to add an enlarged fragment also for E-cadherin similar to Figure 1.
6. Most of the time I met the formula for the fluorescence intensity like this: IF = (intensity in contacts - intensity in the cytoplasm)/intensity in the cytoplasm. Could the authors check this formula as well for their data? Or could the authors explain their choice of formula?
7. How many experiments with microscopy were performed for the fluorescent intensity measurements? There is no information about that.
Reviewer 2 Report
Title: As this study was performed in only one cell type, please specify tissue specificity in the title.
Introduction
· Line 68: Please define PLEKHA7
· Line 70: Please define DROSHA
Materials:
· Please note the potential effects of species variability [human (Collagen IV) vs. rat (Collagen I) vs murine (laminin) vs bovine (fibronectin)] on ECM-cell binding within this study. Collagen post-translational modifications, such as N-glycosylation and hydroxyproline, have species and motif-specificity that may affect efficiency of cell-ECM binding/interaction.
· Please include a schematic of the image analysis and quantification in figure 1 or in the supplemental materials, as it is unclear how this was performed based on the materials and methods section and is pivotal to the manuscript.
· Please include a statistics statement. For example, Figure 6 should be post-hoc corrected, however there is no mention in the figure caption.
Results:
· Figure 2: Please include an apical enlarged merge image.
· Figure 2: The authors should calculate the degree of colocalization between AGO2 and E-cadherin.
· Figure 3: Please see comments above and repeat with DROSHA.
· Please provide rationale ratios used (1:1) for co-ECM studies.
Discussion:
· Please discuss the implication of AP2O2 and DOSHA translocalization.
· The plated alignment of type I collagen fibers here, while unmeasured, is likely random orientation relative to the epithelium. Please discuss the role of collagen fiber alignment relative to epithelium in cancer, cell-ECM binding, and cell adhesion proteins. Additionally, please discuss this limitation of the study.
· According to methods, cells were only cultured for 24 hours post-subculture. While not stated in the methods, it is assumed that cells were treated with trypsin/EDTA for subculturing. Proteomic expression of cell adhesion proteins are known to be dysregulated by proteolytic activity of trypsin in cell subcultures, and not all recover after 24 hours. Please discuss and note this limitation of the study.
Round 2
Reviewer 1 Report
Line 163 - misprint "p120-tatenin"
There is a problem with Figure 1 in pdf file. I can see only the selections but no cells. It looks normal in MS Word file.
Probably, the auhors have downloaded the previous version of the file with original images for blots/gels by mistake. So, I can't check the mentioned file.
Reviewer 2 Report
The authors have made an appropriate effort to address all reviewer comments.